# Weeping in the Face of Fortune: Eco-Alienation in the Niger-Delta Ecopoetics

## Abba A. Abba [1,*] and Nkiru D. Onyemachi [2]

1   Department of English and Literary Studies, Federal University, Lokoja 260101, Nigeria
2   Department of English and Literary Studies, Edwin Clark University, Kiagbodo, Delta State 333105, Nigeria; dorleey@yahoo.com
*   Correspondence: abba.abba@fulokoja.edu.ng

**Abstract:** Scholarship on Niger Delta ecopoetry has concentrated on the economic, socio-political and cultural implications of eco-degradation in the oil-rich Niger Delta region of the South-South in Nigeria, but falls short of addressing the trope of eco-alienation, the sense of separation between people and nature, which seems to be a significant idea in Niger Delta ecopoetics. For sure, literary studies in particular and the Humanities at large have shown considerable interest in the concept of the Anthropocene and the resultant eco-alienation which has dominated contemporary global ecopoetics since the 18th century. In the age of the Anthropocene, human beings deploy their exceptional capabilities to alter nature and its essence, including the ecosystem, which invariably leads to eco-alienation, a sense of breach in the relationship between people and nature. For the Humanities, if this Anthropocentric positioning of humans has brought socio-economic advancement to humans, it has equally eroded human values. This paper thus attempts to show that the anthropocentric positioning of humans at the center of the universe, with its resultant hyper-capitalist greed, is the premise in the discussion of eco-alienation in Tanure Ojaide's *Delta Blues and Home Songs* (1998) and Nnimmo Bassey's *We Thought It Was Oil but It Was Blood* (2002). Arguing that both poetry collections articulate the feeling of disconnect between the inhabitants of the Niger Delta region and the oil wealth in their community, the paper strives to demonstrate that the Niger Delta indigenes, as a result, have been compelled to perceive the oil environment no longer as a source of improved life but as a metaphor for death. Relying on ecocritical discursive strategies, and seeking to further foreground the implication of the Anthropocene in the conception of eco-alienation, the paper demonstrates how poetry, as a humanistic discipline, lives up to its promise as a powerful medium for interrogating the trope of eco-estrangement both in contemporary Niger Delta ecopoetry and in global eco-discourse.

**Keywords:** ecopoetics; eco-alienation; oil politics; Niger Delta poetry; hyper-capitalism; Tanure Ojaide; Nnimmo Bassey; ecocriticism; Anthropocene

## 1. Introduction

This paper analyses Ojaide's *Delta Blues and Home Songs* (1998) and Bassey's *We Thought It Was Oil but It Was Blood* (2002) to understand both poets' common posture that "oil exploration destroys the environment and reduces the opportunity for human survival" (Okuyade 2013, p. 75). Th argument here is that the Anthropocene, the breach between people and nature is implicated in the sense of eco-alienation in the selected poetry collections. Our ecocritical discursive approach will help to foreground how poetry, as a humanistic discipline, lives up to its promise as a powerful medium for interrogating the trope of eco-estrangement both in the contemporary Niger Delta and in global ecopoetics. Cajetan Iheka (2018, p. 7) has argued that "many African societies, despite their complexities and differences, are drawn to an ethics of the earth" in which "certain nonhuman forms, including

animals, plants, and so on, are considered viable life forms worthy of respect". This seems equally so in many other cultures across the world. However, postcolonial Africa has been negatively affected by hyper-capitalist endeavors that dangerously influence the relationship between the people and nature. In his definition of nature, Raymond Williams notes that "nature" "is perhaps the most complex word in the English language" (Williams 1976, p. 184). Although Frisch also believes that "Nature does not need a name" (Frisch 1979, p. 138), the traditional notions of nature see it as the opposite to human culture, technology, history, society, or even metaphysics. As Adam Frank, an American astrophysicist argues in his article titled, "Earth Will Survive: We May Not":

> We often speak of saving the earth as if it were a little bunny. Our earth does not need our saving. What Earth's history does make clear, however, is that if we do not take the right kind of action soon the biosphere will simply move on without us, creating new versions of itself in the changing climate we are generating now (qtd. in Eze 2019, p. 283).

Yet, many recent writers have also observed that the "claims about 'nature' are necessarily mediated by culturally specific prejudices: that when we talk about 'nature,' we are really only talking about our particular society's ideas about nature" (Biro 2005, pp. 3–11). Some eco-critics even go so far as to blame hyper-capitalism for causing the misbalance of social life by allowing commercial interests to influence every aspect of the human experience. In fact, for these scholars, it is an extreme *laissez-faire* capitalism that is marked by greed, selfishness, destruction, wars and exploitation, which is sustained by a deregulation that ensures a free flow of capital with little or no government intervention (Vujnovic 2012, p. 1). For Vujnovic, hyper-capitalism is a Marxist term that depicts a relatively new form of capitalistic social organization, marked by the speed and intensity of global flows that include the exchange of both material and immaterial goods, people and information. Hyper-capitalism in the context of this essay refers to an inordinate quest for economic wealth involving the wanton commodification of the environment.

The urgency to protect nature from human abuse is an idea that is currently embraced by a growing number of Humanities scholars. For Timothy Luke, "many major corporations now feel moved to proclaim how much "every day is Earth Day" in their shop, what a meaningful relationship they have with nature, or why their manufactures are produced with constant care for the planet's biosphere" (Luke 1997, p. 116). In an interesting study on the role of the forest in medieval German literature, Albrecht Classen (2017a, p. 1) illustrates the very important role of the forest in understanding the fundamental relationship between humans and nature already in the Middle Ages. In another related ecocritical examination of the place of water in medieval literature, Classen (2017b) also observes the medieval poets' keen interest in, and exploration of, "the great relevance of water to sustain all life, to provide understanding of life's secrets, to mirror love and to connect the individual with God" (p. 1). Yet, given the advancement in human civilization arising from the alteration of nature for human gain, the question is whether we prefer a form of human relationship with nature that forces us into choosing nature at the expense of everything we derive from its alteration. However, as Biro states:

> In an era in which a widely professed commitment to 'sustainable development' sits rather comfortably beside the voracious consumption of non-renewable resources and a related intensification of stark social inequalities, what seems to be needed is an ecological politics that is 'denaturalized,' or that does not rely on an abstract, reified, 'antisocial' conception of nature (Biro 2005, pp. 197–218).

The central question at this point is what role the Humanities at large and literature in particular play in combating the challenges of the Anthropocene, a notion that positions people at the center of the universe and empowers them to alter nature and the natural environment. Arguing that literature plays a transformative function, Hubert Zapf (Zapf 2016a, p. 139) specifically identifies the ecocritical literary text as engendering "an imaginative counter-force" that is 'indispensable for an adequately complex account of the lives of humans and their place in the more-than-human world' (pp. 108–9). It is also a

kind of discourse that is variously associated with an ecosemiotic force that both emerges from and transgresses traumatizing realities and releases transformative story-telling processes. According to Zapf, the vital interrelatedness between culture and nature has been a special focus of literary culture from its archaic beginnings in myth, ritual and oral story-telling; in legends and fairy tales; in the genres of pastoral literature, nature poetry and in the stories of mutual transformations between human and nonhuman life. As a result, the mutual opening and symbolic reconnection of culture and nature, mind and body and human and nonhuman life in a holistic and yet radically pluralistic way seems to be one significant mode in which literature functions universally and in which literary knowledge is produced (Zapf 2016a, p. 139).

Although far from being a final solution to all human problems, literature proves to be a critical medium to investigate human life in its infinitude of features: highlights, downfalls, losses, successes, happiness, sorrow and so forth (Classen 2020, p. 3). Thus, the signal function of literature is also to question those practices and narratives that seek to destabilize not only people's relationship with nature but also people's relationship with their environment. Iheka also notes that "the distancing of the Africans from their environment was carried into the practice of African literary studies and postcolonialism with an emphasis on the portrayal of rational, modern subjectivities that are often inconsistent with those indigenous practices that connect humans to their environment" (Iheka 2018, p. 10). Iheka's argument is relevant to our understanding of the role of ecocriticism in mediating "ecohuman engagement" (Egya 2012, p. 1) in the Niger Delta ecopoetics. If the deconstructive power of ecocritical discourse symbolically enables it to empower the marginalized, it also enables it to meditate the reconnection of what is culturally subjugated. For instance, the Niger Delta of the South-South in Nigeria is an oil rich region that accounts for the greater percentage of the oil resources in a country that seems to depend solely on oil revenue for its exports. In Michael Watts' 2004 study entitled "Resource Curse? Governmentality, Oil and Power in the Niger-Delta Nigeria," the author observes that Nigeria—the thirteenth largest producer of petroleum and an archetypal oil nation for which petroleum products account for 80 percent of government revenues, 95 percent of export receipts and 90 percent of foreign exchange earnings—currently provides at least five percent of daily US consumption (and over 10 percent of US imports), and West African fields now exceed the volume of US imports from Saudi Arabia (Watts 2004, p. 50). In fact, in a March 2000 presentation to the US Congressional International Relations Committee Sub-committee on Africa, the Petroleum Finance Committee (PFC) foregrounded the great significance of Nigerian oil, taking particular note of the strategic importance of West African oil and the high quality and low cost of its "sweet" reserves—including new off-shore, deep-water discoveries— and demanded substantial foreign investment in the Nigerian oil (Watts 2004, p. 50). However, the region from which much of the resources are generated has been seen as the least developed part of the country, with not only the poorest citizenry but also the most plundered environment as a result of the activities of the local and international oil exploration companies in collaboration with the Nigerian government.

Tanure Ojaide (1998) and Nnimmo Bassey (2000), two important Nigerian poets from the Niger Delta have published lots of poetry collections in which the oil-rich region is perceived as an endangered environment. Unfortunately, the huge and diverse range of scholarship on these poetic works only reflects the economic, social and political implications of eco-degradation in the region but falls short of addressing the trope of eco-alienation, the sense of separation between people and nature, which seems to be a significant trope in Niger Delta ecopoetics. The term "Niger Delta ecopoetics" is used here to refer to the poetic engagements that center on environmental concerns, especially with reference to ecological disruptions in the Niger Delta region of Nigeria. For instance, some scholars have argued that, to a considerable extent, what currently prevails in the Niger Delta oil enclave is a specific variant of internal colonialism and that the exploitative and grossly inequitable entitlement-relations between the Nigerian state and the oil communities explain why "the enormous oil wealth generated is scarcely reflected in the living standard and life chances of the peasant inhabitants of the oil-bearing enclave" (Etteng, qtd. in Ushie 2006, p. 21).

Both Ojaide and Bassey are very prominent poets who have not only deployed their poetic weapons in the fight against environmental injustice in the Niger Delta region, but have also inspired a good number of other poets. They seem to be aware that the struggle for the emancipation of the people of the area should not be left only in the hands of gun-toting militants working in the creeks. The intellectual angle of the war in a way was needed to complement the other forms of activism going on. Thus, beyond being scholar–critics, activists, nationalists, cultural entrepreneurs and novelists, they both, above all, are also prolific, award winning environmentalist poets of Niger Delta extraction. For instance, Ojaide's artistic vision reflects all of this through his long list of creative works, especially his poetry collections, which include: *The Questioner* (2018), *Songs of Myself: A Quartet* (2015), *Love Gifts* (2013), *The Beauty I Have Seen* (2010), *Waiting for the Hatching of a Cockerel* (2008), *The Tale of the Harmattan* (2007), *In the House of Words* (2005), *I Want to Dance and Other Poems* (2003), *In the Kingdom of Songs* (2002), *Invoking the Warrior Spirit: New and Selected Poems* (2000), *When It No Longer Matters Where You Live* (1999), *Invoking the Warrior Spirit* (1999), *Delta Blues and Home Songs* (1998), *Daydream of Ants* (1997), *The Blood of Peace* (1991), *The Fate of Vultures* (1990), *Poems* (1988), *The Endless Song* (1988), *The Eagle's Vision* (1987), *Labyrinths of the Delta* (1986), *Children of Iroko and Other Poems* (1973). His eco-engagement is not limited to poetry. For instance, many of his fictional works comprising short stories and novels also seek to expose and interrogate the universal irony that is embedded in human experience and the Niger Delta eco-disaster. This is the dominant theme works, such as *God's Naked Children* (2018), *Stars of the Long Night* (2012), *The Old Man in a State House and Other Stories* (2012), *Matters of the Moment* (2009), *The Debt-Collector and Other Stories* (2009), *The Activist (A Novel)* (2006), *Sovereign Body (A Novel)* (2004) and *God's Medicine Men and Other Stories* (2004). It is with his poetry that we are concerned in this paper.

Writing in an article entitled "I Want to Be an Oracle: My Poetry and My Generation," published in *World Literature Today* in 1994, Ojaide offers an insight into his creative engagements, which is worth quoting at length:

> My roots thus run deep into the Delta area. Its traditions, folklore, fauna and flora no doubt enriched my *Children of Iroko* and *Labyrinths of the Delta*. This area of constant rains, where we children thought we saw fish fall from the sky in hurricanes, did not remain the same. By the 1960s the rivers had been dredged to take in pontoons or even ships to enter our backyard. Shell B P had started to pollute the rivers, streams, and farmlands with oil and flaring gas. Forest had been cleared by poachers and others to feed the African Timber and Plywood Company in Sapele. Streams and marches dried up. Rubber trees were planted in a frenzy to make money and were soon tapped to death [ . . . ]. To me as a poet, childhood is vital, because it is the repository of memory. That is why the Delta area has been so important to me [ . . . ]. My Delta years have become the touchstone with which I measure the rest of my life. The streams, the fauna, and the flora are symbols I continually tap (Ojaide 1994, p. 15).

Similarly, Ojaide acknowledges the indebtedness of his warrior spirit to his native Uhrobo folk tradition:

> I have been in a warrior spirit in a way; it defines in a sense what I have been doing all along. [ . . . ] [Y]ou see . . . , among the Urhobo people, there is a certain spirit irin [Ivwri] and it's like either a shrine or some things [thing] built during the slave trading era and was supposed to be imbued with the power to escape being captured and also; recapturing those who had been captured by raiders. In a way, it has, to me, become the symbol of existence (Kalu 2000, p. 13).

Niyi Osundare (Osundare 2002, p. 10) eulogizes Ojaide's generation as one historically positioned to "match accessible, elegant style with relevant content". Yet, Ojo Olorunleke (2017, p. 12) contends that Ojaide's major thrust in his poetry, especially in *The Fate of Vultures* and *Delta Blues and Home Songs*, seems ideological given their political, socio-economic and cultural undertones. Moreover, identifying

the politico-cultural imagination in Ojaide's poetry, Olafioye T. (Olafioye 2000) describes Ojaide's "favourite motif", his "usual political humdrum" (138) as "the harangue of the state or rulership for corruptive practices" (94). For Olorunleke, Ojaide's poetry offers a diagnosis of the contradictions of the operators of the Nigerian State and their state policies at different historical moments in Nigeria's existence. Arguing that Ojaide's poetry belongs to the post-Independence period of the Nigerian Second Republic and after, Olorunleke points out that *The Fate of Vultures* and *Delta Blues and Home Songs,* for instance, dwell on the distortions brought about by oil exploration, which has afflicted the nation's well-being:

> Placed on a historical continuum, Ojaide's poems provide vistas on the political actors of Nigeria's Second Republic of the Shagari era, through General Buhari's draconian brief spell, General Babangida's windy years of the implementation of an International Monetary Fund-inspired Structural Adjustment Programme (SAP), which effectively disarticulated Nigeria's sub-structural economic foundation and inaugurated the political experiment of "new-breed" politicians, and the inglorious days of the maximum ruler, General Sani Abacha (Olorunleke 2017, p. 12).

Sule Egya (2018, p. 1) also avers that the dependence of Ojaide's poetry on orality implies its rootedness in nature and that, more crucially, "nature in his poetry is not merely evoked as an aesthetic strategy—an embellishment of what many have regarded as an overwhelming political theme in his poetry. Nature (the natural environment, biodiversity, flora and fauna) is also thematized as home—now a lost home in the face of modernity and petrodollar global capitalism."

Retracing the tracks forged by Ojaide in his career as poet, environmental activist and architect, Nnimmo Bassey also attempts to expose the commodification of socio-economic relations and ecological dissonance in the Niger Delta (Nwagbara 2010, 2012, p. 61). An indigene of Ibibio in Calabar, Nigeria, he went into human rights activism in the 1980s and in 1993 he became a co-founder of the Environmental Rights Action (ERA), also known as Friends of the Earth Nigeria (the national chapter of Friends of the Earth International (FOEI), the world's largest grassroots environmental network), to confront environmental human rights issues. Bassey's major focus in his campaign is oil, and the huge destructions experienced by communities and other countries where oil is produced. In 2010, he won the prestigious *The Right Livelihood Award* "for revealing the full ecological and human horrors of oil production and for his inspired work to strengthen the environmental movement in Nigeria and globally" (Friends of the Earth Foundation 2016, p. 2). In his writing career, which, for the most part, cuts across poetry, environmental activism and architecture, Bassey primarily interrogates environmental injustice across the globe. Some of these works include: *Patriots and Cockroaches* (1992), *Beyond Simple Lines: the Architecture of Chief G.Y. Aduku and Archcon (with Okechukwu Nwaeze)* (1993), *The Management of Construction* (1994), *Poems on the Run* (1994), *Oilwatching in South America* (1997), *and Intercepted* (1998), *We Thought It Was Oil but It Was Blood* (2002), *Genetically Modified Organisms: the African Challenge* (2004), *The Nigerian Environment and the Rule of Law, ed* (2009), *Knee Deep in Crude* (2009), *To Cook a Continent: Destructive Extraction and Climate Change in Africa* (2011), *I Will not Dance to Your Beat* (2011). In fact, Osundare (Osundare 2002) asserts that "Bassey's poems distil meaningful music from the adversity of imperilled freedom." Patrick Naagbanton (Naagbanton 2018, p. 4) also notes that Bassey is a serious poet with a serious message, a message of life and death, respect for human rights, environmental conservation and protection. Bassey's poetry does not spare the oil and gas transnational corporations, governments and powerful individuals who threaten the people's future. In a piece written to celebrate Bassey's 60th birthday, Naagbanton describes him as "our travelling, weeping and protest poet on the run" (n.p.).

To be sure, like Ojaide, Bassey acknowledges the influence of Ken Saro-Wiwa, the great Niger Delta playwright, poet, novelist, newspaper columnist, short story writer, activist and environmentalist in his activist consciousness. In a foreword to the book, *Silence Would Be Treason—Last Writings of Ken Saro-Wiwa,* Bassey writes:

Ken Saro-Wiwa and Sister McCarren (the Irish revered sister of the Catholic Church) both influenced my life and growth as an environmental justice advocate. In addition, Saro-Wiwa challenged me as a fledging writer who thought I would find a niche as a poet and short story writer. His pioneering work in building a virile environmental justice movement as well as the rights of minorities in Nigeria remains outstanding and continues to inspire campaigners around the world. I recall his visit to my humble home in Benin City when he came to lead a conference of the Association of Nigerian Authors (ANA) in 1994. It was a memorable occasion and I heartily drank from his spring of wisdom on a variety of topics (Bassey 2013, "Foreword").

After his annulment of the 12 June 1993 presidential election which threw Nigeria into a serious political turmoil, General Ibrahim Babangida, the then Head of State, formed his ineffectual Interim National Government (ING) and placed one of his loyalists, Ernest Shonekan, at the head before he fled the country in August the same year. However, the interim government lasted only for three months before it was overthrown by Shonekan's brutal Defense Minister, General Sani Abacha, who took over as head of the new junta. Abacha's reign of terror led to the harassment, imprisonment and, most often, execution of those who complained about his tyrannical and corrupt regime. It was during this period that Ken Saro-Wiwa was hanged alongside other environmental rights activists in the Niger Delta. It was for this reason that Bassey, who was openly and deeply involved in activism, had to go into hiding in Ibadan, the Oyo State capital, for a period of over four months. While writing from his hideout, his family members were harassed, intimidated and detained without trial. However, luck ran out for the poet on Wednesday, 5 June 1996 when the rampaging operatives of the State Security Service (SSS) kidnapped him on his way to Accra, the capital of Ghana, to attend the World Environment Day. However, all the intimidation and torture could not compel him to run away from his country; he was on the run within. One of Nigeria's renowned poets, Odia Ofeimun observes that Bassey's poem, written from his hiding place in Ibadan and titled, "On the Run" essentially "confronts the banal misuse of power as well as the anger, riding from Goma to Ogoni, and Freetown and Banjul to Abuja, were as prefigured in the poem, "Shell . . . " (Naagbanton 2018, p. 5). Philip Aghoghovwia (Aghoghovwia 2014, p. 42) also illuminates Bassey's uncommon commitment to the fight against environmental injustice:

> Bassey seems to testify to how collective responsibility on behalf of nature might be a most effective way to wrestle the planet from corporate greed. . . . He writes with an insurrectionary fervour as if he were addressing a gathering as on a street live demonstration. Here is a poet with an abiding commitment to the politics of non-silence.

Bassey's collaboration with national and international organizations makes him one of Africa's leading voices against eco-devastation by multi-national corporations who equally ignore the rights of the local population. Interestingly, he and Ojaide share a common poetic vision about the disconnect in eco–human relationships. Specifically, the selections of their poetry analysed herein demonstrate that, although evil commonly begets evil, often goodness also bears the seed of evil.

## 2. Weeping in the Face of Fortune

Ogaga Okuyade (Okuyade 2016, p. 1522) has posited that "we are not only in an age of global environmental catastrophe nor do we only live in a watershed moment; the twenty-first century person is in a perpetual state of crises, a situation created by mankind's uncanny translation of the functions of the environment to meet his/her ever insatiate greed for satisfying his/her unquenchable taste for resources." From this dimension, the earth is taken as a mere product to be consumed. Ogaga's position equally provides an important background for Iheka's ecocritical insights in his newly published monograph entitled, *Naturalizing Africa*. According to Iheka:

> The anthropocentric position, under which ecological violence has become normalized, places humans at the center of the universe and sanctions those activities meant to "prosper" the

human, even when they are detrimental to other beings; but the decentering of the category of the human counters that exceptionality. If human beings retain any exception here, it is their unique capacity to significantly alter the ecosystem, for better or worse in the age of the Anthropocene (Iheka 2018, p. 13).

How do we understand the Anthropocene and its relationship to our literary and humanistic discourse? As Reinhold Münster (forthcoming, p. 2) has noted, some geological hypotheses today expand the range of questions which could be addressed by both the natural scientists and the scholars in the Humanities. The Anthropocene is one of such issues. Our world is increasingly fragmented into natural and human components in which the human being, as the sole authority bestowing meaning to history, loses its relevance and then even also its own existence (Frisch 1979, p. 138). It is a concept required to explain and comprehend the consequences of human attempts to conquer the entire earth. Paul J. Crutzen identifies the latest period in the history of humankind and the earth as "Anthropocene", which began in the time of the late European Enlightenment and the period of early industrialization. It is the period that follows the age of the Holocene in which nature with its essence was dominant:

> The beginning of the Anthropocene can be dated to the late eighteenth century since investigations of air bubbles within the ice core drillings revealed that the concentration of carbon dioxide and methane within the atmosphere began to grow at that time all over the world. This date coincides with James Watt's invention of the parallelogram in the year 1784, together with a decisive improvement of the steam engine (Crutzen 2019, p. 171; quoted from Münster forthcoming).

In the Anthropocentric age, "the human being dangerously exerts a force which unstoppably changes the earth and, in the near future, also other planets within the solar system" (Kaku 2019, pp. 81–90). We see that the exploration of space, the economically driven flights to the moon and the asteroids, as well as the mining of the deep layers of the earth's crust and the depth of the oceans, indicate that today, human society transforms the earth according to its own concepts.

Albrecht Classen observes that "in face of the imminent climate crisis, global warming, migration, and many other problems, technology and sciences have been able to provide only partial answers, or have even been the cause of catastrophic developments (nuclear power, loss of the ozone layer in the atmosphere, warming of the world oceans, environmental pollution; military aggression as a result of new weaponry)" (Classen 2020, p. 2). As people alter the earth and push the small remains of wilderness to the margin (Schama 1995), we find that "we live on a badly damaged earth and are in the process of making our own home planet unlivable" (Schmelzer and Vetter 2019). Eva Horn also illustrates how people create an environment that is now determined by an uncontrollable dynamic of its own:

> The Anthropocene is not simply a crisis that will pass some day in the future; instead, it is an absolute fracture; a breaking down, if not collapse, of the previously stable ecological conditions of the Holocene, which include: Environmental conditions under which everything came into being what we know as human civilization, such as settlements, agriculture, cities, trade, complex social institutions, tools and machines, but also the written culture and all other media serving to record, pass on, and interconnect knowledge. If the Holocene was the cradle of civilization, the question arises what the destruction of those conditions—the social organization, technology, and the relationship of people with themselves and the world—will mean. The Anthropocene stands for a future the outlines of which we are only beginning to grasp (Horn and Bergthaller 2019, p. 11).

The obvious result of the Anthropocene is the sense of eco-alienation, which is the sense of human separation from nature and more natural essence. Albert Bergesen (2012, p. 1) argues that eco-alienation in the 21st century is the successor to two earlier ages of alienation: the religious

alienation, which refers to the sense of separation from God; the Renaissance age of human alienation, concerned with human separation from the self and society.

The Renaissance notion of humanity perceives humans as the measure of all things, a conception that makes them cut-off from their animal nature. Ironically, if this constituted a huge advance, it also generated a new alienation, as there is now a growing awareness of a new breach between human beings and nature. In the midst of a philosophical struggle to find a new conceptualization for our existence, it is important to realize that, as part of nature, humans are eco-beings first, and human-beings second. Iheka thus advocates a change of human perspective in order to see and relate to the plants and animals and the lands and forests around us as constitutive of the living world and not as mere resources for indiscriminate exploitation. Anna Peterson equally notes that the belief in the human–nonhuman relationship in most cultural spaces can "inspire and legitimize efforts to preserve the delicate web of social and natural relationships in which we all exist" (Peterson 2018, p. 26).

Literature and the Humanities come in here to play the role of bridging the gulf between human beings and nature. It is within the foregoing theoretical context that we locate the representation of eco-alienation in Ojaide's *Delta Blues and Home Songs* and Nnimmo Bassey's *We Thought It Was Oil but It was Blood*. Both poets translate their ecopoetics into an imaginative counter-discourse to probe and subvert the fracturing experiences in their region. Seeking to foreground the necessity for man's reconnection with the natural world, the value of their texts is primarily located in their demonstrable attempt to underscore the need to preserve the integrity, stability and beauty of the eco–human relationship.

The eponymous poem in Ojaide's poetry collection entitled "Delta Blues" is generally an ecopoetic lamentation of the tragedy of eco–human disorder. Dedicated to Ken Saro-Wiwa and other civil-rights activists, it articulates the urgency for environmental activism, keeping in constant relief the bloody sacrifices that have been made in the struggle. It is important to note that the poem alludes to the hanging of Ken Saro-Wiwa in 1995, along with some other minority rights activists in Ogoni-land for resisting environmental degradation in the region by the Sani Abacha-led Nigerian government. The reference to that episode to a large extent supports the poetic claim that the Niger Delta indigenes now perceive oil wealth in their region as a source of death rather than a preserver of human life. In this section of his poetry, the poet–speaker illuminates the notion of oil wealth in the Niger Delta as a metaphor for death:

> This share of paradise, the delta of my birth,
> reels from an immeasurable wound.
> Barrels of alchemical draughts flow
> from this hurt to the unquestioning world
> that lights up its life in a blind trust.
> The inheritance I sat on for centuries
> now crushes my body and soul . . .
> My nativity gives immortal pain
> masked in barrels of oil (p. 15)

The poet laments that his place of birth, hitherto a paradise, that has since then produced barrels of oil, has been turned into one dangerous environment that threatens the existence of its inhabitants. We encounter here the poet's constant awareness that oil, thought to sustain human life, is now what threatens human life. The attraction of oil as an economically viable liquid substance makes it an oxymoronic element encompassing both life and death. It has become a mask of immortal pain.

While the lament endures, albeit uncontrollably, the poet painfully draws accusatory attention to the hyper-capitalists—the perpetrators whom he describes as "a cabal of brokers":

> I stew in the womb of fortune.
> I live in the deathbed
> prepared by a cabal of brokers

> breaking the peace of centuries
> and tainting not only a thousand rivers,
> my lifeblood from the beginning,
> but scorching their sacred soil was debauched
> by prospectors, money-mongers. (p. 16)

This cabal is accountable for the violence that rocks the land, depriving the people of their age-long peace. Their activities have not only poisoned many rivers, which have endured over the centuries as stores for natural resources, they have also destroyed the cultural ecology, "the sacred soil". This, for the poet, is a clear example of the monstrosity of hyper-capitalists in the region.

In "Wails", another poem in *Delta Blues and Home Songs*, the poet invokes the deity "Aridon" to grant him the voice to draw the world's attention to their plight because the country has become a devourer of its own. Aridon is one of the gods in the pantheon of Urhobo to whom Ojaide alludes, through which he establishes the inter-connectedness between humans and the gods. Rather than eulogizing and paying tributes to Greco–Roman or European classical gods, Ojaide looks up to his native Urhobo gods for inspiration. Aridon occupies an exalted position in his poetry as the god of memory and remembrance, whom he depends on for inspiration and the extra edge needed to perfect his art (Ojaruega 2015, p. 154). As Ojaruega further notes, Aridon and Uhaghwa are sometimes used interchangeably to denote the god of memory or flawless performance. While both deities appear to be foremost in the poet's mind, he acknowledges deities in the Urhobo pantheon in his poetry, such as: Eni, the god of truth; Ivwri (also spelt Iphri), the god of restitution; Mami Wata, used interchangeably with Olokun, the god of good fortune, wealth and beauty (Ojaruega 2015, p. 155). Despite his devotion to these deities, the poet still believes in the Supreme God, known in Urhobo as Osonobrughwe or Oromowho, the Great Creator in his tradition, whom he prays to for the good of society, the nation, and the self. In this section of his poetry, he beckons Aridon for the inspiration to rouse his people into action through his art:

> Aridon, give me the voice
> to raise this wail
> beyond high walls.
> In one year, I have seen
> my forest of friends cut down,
> now dust taunts my memory . . .
> I must raise the loud wail
> so that each will reflect his fate . . .
> The boa thoughtlessly devours
> its own offspring, Nigeria's
> A boa constrictor in the world map (p. 18).

This is a subversive voice aimed at countering the activities of those who, through their greed, create a sense of alienation between man and ecology. In our age "man suffers not only from war, persecution, famine and ruin, but from inner problems, a conviction of isolation, randomness, meaninglessness in his way of existence" (Saleem 2014, p. 67). The poem's mournful temper reaches its passionate intensity when the poet reflects on the mowing down of his friends and environmental rights activists. For the poet, to slaughter one's own offspring is to be caught up in a tragedy of self-annihilation. Pointing to the horrible trappings of hyper-capitalist enterprise in the region, he insists that the Nigerian State has transformed into a "boa constrictor" that thoughtlessly devours its own offspring. To be sure, "the growth of the personality of man and the factors responsible for alienation are subject to the influence of social-conditions on human existence" (Fromm 1866, p. 10).

Turning its gaze back home, the poet appropriates some significant cultural idioms to berate the selfish and shameful activities of the elders of the region who work in complicity with the hyper-capitalist commercialists in wrecking the region. Painting a picture of a landscape suffocating

under the weight of sorrow in the poem "On the World Summit for Children at the UN 1990", the poet illuminates the tragic tension that surrounds the people and their environment. For him, "the more than man" is in complicity with human wilfulness in the smothering condition. He likens the difficulty of capturing these sorrows in words to the unfortunate history that forbids the dog from shedding tears, and the goat from sweating:

> Dogs will never shed enough tears
> to tell their sorrow,
> goats will never sweat enough in a rack
> to show the world their desperation. (p. 17)

While the eco–human apocalypse rages on, "babies suffocate from the game/of loveless elders of state … " (p. 70) who, while posing as redeemers, defraud their own people and still pretend that they are fighting on their behalf. Ojaide tactfully brings them into the poem to lampoon their monstrous nature:

> We are in league with powers
> To wreck one vision
> With lust for more visions
> To refashion a proud world—
> With the same hands … (p. 15).

The poet seeks a counterforce to the massive exploitation supervised by the elders of the land, whose hands destroy the land while pretending to refashion a new world. Ojaide's poetry thus offers a virulent attack on the greed and desperation that subject eco-humanity to "omnipresent destruction" (Zapf 2016b, p. 512).

Like Ojaide's *Delta Blues and Home Songs,* examined above, Bassey's *We Thought It Was Oil but It Was Blood* offers a counter-discursive ideation on the destruction of the cultural ecology in the Niger Delta. In fact, Philip Aghoghovwia has argued that Bassey provides us with a counter-narrative to the hegemonic narratives which political gatherings on account of climate change engender in the public sphere (2014, p. 61). Bassey's eco-activism in *We Thought It Was Oil but It Was Blood* challenges the devastation that oil extraction has brought to the people and critiques the ways in which the trajectory of oil permeates every aspect of being and dominates the quotidian for the inhabitants of this environment. Reacting to why he chose poetry as a trench for his eco-activism, Bassey noted:

> I found that in the struggle it's essential to take some aspects of performance. In the African context, a lot of social struggles are carried on the vehicle of poetry and song. And so, I began writing poetry seriously in the early 1990s [ … ]. I found poetry to be very useful in terms of mobilizing resistance, getting people to feel a part of the movement and so some of my poems are not just for people to read quietly, but for people to be part of the reading so that there are calls and responses; so, for example, when I say 'we thought it was oil' the audience responds 'but it was blood' (in Aghoghovwia 2014, pp. 39–40).

Through the appropriation of songs as a cultural element in the mobilization of resistance, Bassey draws attention to the significance of music in mitigating the pain of eco–human tragedy in the poem. For Jeffrey Andrew Barash, all this represents the collective memory that creates the basis of our lives and makes possible the growth into the future (Barash 2016). In the funeral traditions across many African cultures, there is always a symbolic gesture to mitigate the terror of death with music and dance. In an interview with Oyeniyi Okunoye, Ojaide also underscores the influence of his native myths, songs and dance-songs in his own poetry:

> The songs of my Urhobo people made an impact on me, especially the Udje dance songs. Also, the folklores and the myths had an impact on me in their didactic content. The rural environment which enables me to follow my grand- father and uncles to farm and fish also

affected me with its fauna and flora. So, the entire traditional village life has a lasting impact on me (Okunoye 2002, p. 223).

Despite the attempt to subvert the exploitative oil activities in his poetry, Bassey also appropriates songs and dance as a surrealistic idiom to transform the sobriety of eco–human alienation to "a rousing affirmation of the enduring value of human life" (Diala 2015, p. 97). In the eponymous poem that introduces the entire sequence of Bassey's poetry collection, entitled "We thought it Was Oil", the poet pictures his people in a dance—a cultural performance through which they celebrate the gift of oil wealth. The rhythmic movement of their waists speaks well of joy and fulfilment, announcing a triumph over poverty. But these thoughts suddenly fizzle away as they behold before them a delusory mask of death:

> The other day
> We danced on the street
> Joy in our hearts
> We thought we were free
> Three young folks fell to our right
> Countless more fell to our left
> Looking up,
> Far from the crowd
> We beheld Red hot guns.
> We thought it was oil
> But it was blood. (p. 12)

Similar to the ironic situation in Ojaide's *Delta Blues and Home Songs*, oil in *We Thought It Was Oil but It Was Blood* is also envisioned as an oxymoronic element that contains within itself the seed of joy and pain. Much like the Dionysian element often present in Greek tragedy, oil is seen as terror, which is likely to turn on its wielder and tear him apart. If indeed it is a source of joy, it is also a source of disillusionment; if it is a blessing, it is equally a curse; if it is a source of life, it also bears the seed of death. Keniston points out that "most usages of alienation share the assumption that some relationship or connection that once existed that is 'natural', desirable or good, has been lost" (Keniston 1965, p. 5). There is a disruption, tension and "a psychological phenomenon, an internal conflict, a hostility felt towards something seemingly outside oneself which is linked to oneself, a barrier erected which is actually no defense but an impoverishment of oneself" (Finkelstein 1965, p. 7). This conflictual image of oil constructs a sense of alienation between people and nature. The refrain "We thought it was oil/ But it was blood" underscores the painful irony:

> Heart jumping
> Into our mouths Floating on
> Emotions dry wells
> We leapt with fury
> Knowing it wasn't funny
> Then we beheld
> Bright red pools (p. 13)

Depicting a sense of uncertainty and an atmosphere of terror which haunts the margins of the poem, the poet calls to memory those who have been smothered in the struggle to rid the land of the economic rapists. The tone of subversion and resistance is perhaps provoked by the monstrosity of human greed, symbolized by the "bright red pools". The metaphor of "emotions drying up the wells" demonstrates the degree of consternation that has overcome the poet's community when they saw that the enemies were armed with "bright red pistols." Many lives have been consumed by the corporate lawlessness:

> First it was the Ogonis
> Today it is Ijaws
> Who will be slain this next day?
> We see open mouths
> But hear no screams
> Tears don't flow
> When you are scarred
> We stand in pools
> Up to our knees
> We thought it was oil
> But it was blood [ … ]. (p. 13)

The inferno of disintegration rages beyond clannish boundaries as it takes its toll on all, moving from the Ogonis to the Ijaws. It moves with a trail of anxiety as nobody knows who will be killed next. Like wildfire, its flaming tongue is all-consuming. There is an omnipresence of terror and death in the corporate bid to forcefully protect commercial interest:

> Dried tear bags
> Polluted streams
> Things are real
> When found in dreams
> We see their Shells
> Behind military shields
> Evil, horrible, gallows called oilrigs
> Drilling our souls
> We thought it was oil
> But it was blood [ … ]. (p. 14)

The oil companies, shielded by government-sponsored military force, carry on with their activities, ignore the awful experiences of the people arising from the oil spillage and the air pollution and other ecological hazards that characterize these activities. The oil rigs are pictured as "evil, horrible gallows, drilling our souls," a description that suggests the idea that, to the people living in the oil region, the murderous international oil hunters, in collaboration with the insensitive Nigerian government, have transformed the environment into a slaughter site.

However, the poet reveals that they have come to a creative moment in which the exploited must overcome fear and struggle to regain his voice through subversion. According to Deleuze, a subject may seem fragmented and temporarily disunited, but he may develop a certain amount of consistency that comes from the continuing power of relocation (qtd. in Sunil Kumar 2015, p. 68). The fragmented subversive voice in Bassey's poem now takes a bold resolve:

> They may kill all
> But the blood will speak
> They may gain all
> But the soil will RISE
> We may die
> And yet stay alive
> Placed on the slab
> Slaughtered by the day
> We are the living
> Long sacrificed
> We thought it was oil
> But it was blood [ … ] (p. 14)

In what approximates to a biblical vision about those who can kill the body but are not able to kill the soul, he avers that though "they may kill all/But the blood will speak/They may gain all/But the soil will RISE." The oxymoronic vision that "We may die/And yet stay alive" suggests that no degree of subjugation can stop the free flow of subversion. He demonstrates the spirit of defiance by which the historical hero has been known and exalted.

In "When the earth bleeds," the poet ridicules the logic of petro-modernity, with the help of a critique of modernity's fake promise, to alleviate the suffering that defines existence in the region. He suggests that petro-capitalism has negatively impacted the human ecology, and that its logic of progress through oil exploration and development creates violence and death:

> I hear that oil
> Makes things move
> In reality check
> Oil makes life stop
> Because
> The oil only flows
> When the earth bleed (pp. 14–15)

Enumerating various kinds of eco-disasters in the region, he reformulates oil as a metaphor for human blood, which flows only when the earth bleeds:

> A thousand explosions in the belly of the earth
> Bleeding rigs, bursting pipes
> This oil flows
> From the earth's sickbed
> Because
> The oil only flows
> When the earth bleeds (p. 15)

He suggests that the oil companies are evil and must be stopped from their dark operations. For this, he threatens to expose their excessive greed:

> They work in the dark
> We must lift up the light
> Quench their gas flares
> Expose their greed
> Because
> The oil only flows
> When the earth bleeds. (p. 16)

The call here is to aggressive confrontation against the exploitative activities of the oil corporations. The poet beckons on his people not to be silent but to rise and expose the greedy corporations and stop them from causing further damage to their environment.

The poet now observes that, while people are dying at home due to the dangerous activities of these companies, the communities' political representatives are busy with empty talks at conferences where their plight should have been receiving the necessary attention. But those political talks yield nothing after all:

> In conference halls
> We talk in gardens of stones
> The ocean waves bathe our eyes
> But in Ogoniland we can't even breathe
> Because
> The oil only flows

> When the earth bleeds
> What shall we do?
> What must we do?
> Do we just sit?
> Wail and mope?
> Arise people, Arise
> Let's unite
> With our fists
> Let's bandage the earth
> Because
> The oil only flows
> When the earth bleeds
> The oil only flows
> When the earth bleeds. (pp. 16–17)

Just as Ojaide invites Aridon to help in fighting the destroyers of the eco–human interaction, Bassey also invites his kinsmen to arise and fight the menace. This is an invitation to violence because the poet feels that diplomacy has failed, making a violent confrontation inevitable. Both poets suggest that what is needed to wrest human ecology from the clutches of corporate greed is a collective responsibility and violent activism, rather than passive lamentation where "we just sit", "wail" and "mope". In fact, mere oral complaints against environmental devastations are not strong enough to arrest the tide of petro-imperialism.

## 3. Conclusions

Corporate greed in oil exploration, involving a collaboration between the Nigerian government and international oil exploration companies in the Niger Delta, has led to experiences that make the inhabitants perceive oil wealth as the people's enemy. The quest by humans to alter the earth or transform nature creates room for hyper-capitalist activities in which human values, ethics, and morality are suspended. Unfortunately, human society can hardly survive in the absence of these ideals. In their attempt to address this situation and bring about desirable changes in society, literature, in particular, and the Humanities at large interrogate the implication of the Anthropocene in the origination and sustenance of eco-alienation in society. In one of his arguments, Ojaide posits that society cannot prosper without stable institutions—one of which is literature (Ojaide 1994, p. 15). Although the positive transformation of nature could bring about social advancement, there is a need for "denaturalized" ecological politics that reject the antisocial conception of nature as it engenders a sense of separation between humans and nature. Thus, in their meditation on the conflicting relationship between man and ecology, Ojaide and Bassey develop their ecopoetry as imaginative counter-discourse, not only for exposing the corporate greed that characterizes hyper-capitalist activities in the Niger Delta region, but also for exploring the pains and sense of eco–human alienation it engenders. They articulate the tragic irony of a people blessed with oil wealth but who, instead of enjoying fulfilling existence, are daily confronted with the omnipresence of death. Observing that both Ojaide and Bassey have made audacious attempts to interrogate the Anthropocene at large and hyper-capitalism in particular, and how both concepts breed eco–human alienation in their ecopoetry, the paper suggests in its conclusion the urgency to forestall any activity that threatens eco–human order. This is important especially in the present era of global climate change in which humankind is facing many challenges all over the world arising from environmental devastation. This way, Niger Delta ecopoetics firmly establishes the important role of literature and the Humanities as powerful media for articulating and negotiating the human condition in global eco-discourse.

**Author Contributions:** A.A.A.—Conceptualization, methodology, formal analysis, writing original draft and proof-reading. N.D.O.—investigation and preparation of manuscript. All authors have read and agreed to the published version of this manuscript.

**Funding:** This research received no external funding.

**Acknowledgments:** We are grateful to the DAAD (German Academic Exchange Service) and the Institute for Asian and African Studies, Humboldt University, Berlin, for a Postdoc fellowship during which period part of this research was undertaken. Our gratitude also goes to Albrecht Classen, University Distinguished Professor and Director of Undergraduate Studies, Dept. of German Studies at the University of Arizona for bringing out what is best in this paper through his insightful critiques, painstaking suggestions and uncommon thoroughness.

**Conflicts of Interest:** The authors declare no conflict of interest.

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
