# Peer review of "Weeping in the Face of Fortune: Eco-Alienation in the Niger-Delta Ecopoetics"

_humanities, doi:10.3390/h9030054_

Round 1

Reviewer 1 Report

I was deeply impressed - in fact moved in places - by the sophistication, articulateness and insight of this study of Niger-Delta ecopoetics. The author has a commanding grasp of the socio-economic and geopolitical forces at play in these contexts, and they use this to enhance their readings.

The interpretations of the works of Ojaide and Bassey are subtle and nuanced. The reading of the 'alchemical draughts' in 'Delta Blues' as a mirage that returns as a circular violence seems particularly strong. Likewise, the observations about voice in relation to the 'Aridon' are highly pertinent to how this poet introduces a destabilising, disruptive polyvocality, or what the author calls in a memorable phrase: "subversive voice".

I was intrigued to see how this "subversive voice" also emerged in the analysis of the excerpt "we are in league with powers"... this sense of speaking "in league with" seems essential as it might also constitute a subversion of some eco poetic buzz words such as "sympoeisis". What happens to ideas of "sympoeisis", such as in Haraway's Staying with the Trouble when power dynamics, infrastructures and the economic factors involved in oil extraction are taken into consideration? Without needing to overtly make this critique, your analysis of voice and speaking 'in league with' opens up this critical possibility, and you build on this when your essay moves on to consider Bassey.

I was particularly impressed here with your thoughtfulness around 'counter-narrative'. I found myself thinking of the work of Édouard Glissant in Poetics of Relation here, the musical elements in Bassey's work operating as "flashpoints" within the poetic surface. 

The conclusion is sound and comprehensive, but in a sense, your essay feels like it is in important ways an opening up of rather than a closing down of, this analysis. I was inspired by this essay to look more in depth at the work of these poets and in these contexts with renewed urgency.  

Author Response

No suggested revision has been made by this reviewer.

Reviewer 2 Report

The title of the second chapter should not be "Results/discussion" but something else, like "analysis" or "study of the materials".

The notes are incomplete, there should always be name, year and page.

Some more contextual information would be valuable for an average reader. For instance, some words about the oil business in Nigeria and a brief presentation of Ken Saro Wiwa, in a note.

The term "hypercapitalism" should be briefly explained, and "ecopoetry" as well. It seems to be a universal trend but this is not told to the reader.

It is perhaps too generalizing to speak about "African" belief, worldview etc. (97-99).

The research method is not presented, but it is some kind of analytical description of the content.

Author Response

Reviewer's comments:

1. Treated. Subtitle changed to 'Analysis'

2. Treated. Notes and in-text references completed now in the 'name, year and page' order.

3. Noted with gratitude but since the article does not centre on Saro Wiwa, we hope that the available information on him should suffice. As we have noted in the article;

Dedicated to Ken Saro Wiwa, and other civil-rights activists, it articulates the urgency for environmental activism, keeping in constant relief the bloody sacrifices that have been made in the struggle. It is important to note that the poem alludes to the hanging in 1995 of Ken Saro-Wiwa with some other minority rights activists in Ogoni-land for resisting the environmental degradation in the region by the Sani Abacha-led Nigerian government. The reference to that episode to a large extent supports the poetic claim that the oil communities and oil wealth in their region have become strange bedfellows.

This, we believe offers the passing insight on Saro Wiwa in the text.

4. Treated. The terms "hypercapitalism" and "ecopoetry" have been briefly explained.

5. Treated. "African" belief, worldview etc. (97-99). "most African cultures' now used to eliminate the sense of generalisation.

6. Treated. The analytic method used has been clearly stated.